# Mobile Apps for Patients with Peritoneal Dialysis: Systematic App Search and Evaluation

**DOI:** 10.3390/healthcare12070719

**Published:** 2024-03-25

**Authors:** Shu-Mei Chao, Ming-Ling Wang, Yu-Wen Fang, Mei-Ling Lin, Shu-Fen Chen

**Affiliations:** 1Department of Nursing, School of Nursing, Tzu Chi University of Science and Technology, Hualien 970302, Taiwan; shumei@ems.tcust.edu.tw (S.-M.C.); lin36@ems.tcust.edu.tw (M.-L.L.); 2International Ph.D. Program in Gerontology and Long-Term Care, School of Nursing, Taipei Medical University, Taipei 110301, Taiwan; aileenww@tmu.edu.tw; 3Department of Nursing, Shuang Ho Hospital, Taipei Medical University, New Taipei City 235041, Taiwan

**Keywords:** mobile apps, peritoneal dialysis, systematic app search, mHealth

## Abstract

Background: Peritoneal dialysis (PD) is one type of renal replacement therapy. If patients have problems during the dialysis process, healthcare providers may not be able assist the patients immediately. mHealth can provide patients with information and help them to solve problems in real-time, potentially increasing their willingness to choose PD. Objective: The objectives of this study were to conduct a comprehensive review of free mobile applications for patients with PD on the Internet and to recommend suitable mobile applications to facilitate patient self-management and health. Methods: We conducted a systematic search for PD mobile applications on Google Play and the Apple iTunes Store from 3 to 16 June 2023. Results: A total of 828 identifiable mobile applications were initially identified, and ultimately, 21 met the inclusion criteria. The Mobile App Rating Scale (MARS) assessment of the applications revealed the highest score in the functionality domain, followed by the aesthetics, information, app-specific, subjective quality, and engagement domains, respectively. In the comprehensive self-management of PD, the highest percentage was related to disease-related information. Conclusion: The findings of this study suggest that some applications, with the highest quality, can be recommended to patients for use in English or traditional Chinese.

## 1. Introduction

The global prevalence of chronic kidney disease (CKD) is approximately 13.4% [1], and in Taiwan, it is 12% [2]. When patients with CKD progress to end-stage renal disease, renal replacement therapy, such as hemodialysis (HD), peritoneal dialysis (PD), or kidney transplantation, may be needed to replace kidney function [3]. Patients communicate with healthcare professionals (HCPs) upon receiving HD twice or thrice a week; therefore, they have a greater sense of security. When patients are undergoing HD, HCPs educate them on how to deal with discomfort and physical symptoms and clarify concerns. Patients perform PD procedures either by themselves or with their caregivers’ help; they have clinical appointments once or twice monthly with respective HCPs [4,5,6]. Therefore, they may not immediately solve or relieve their dialysis-related problems, symptoms, or psychosocial disturbances such as depression; they cannot communicate and discuss their conditions with an HCP in time. Additionally, peritonitis occasionally occurs [7]. The above factors affect the adoption of HD instead of PD, particularly in Taiwan [6,8]. There are some advantages when patients adopt PD; they have more time flexibility and independence [5] as well as reduced medical expenses [9]. Mobile health (mHealth) continually monitors reminders, tracks patients’ conditions, enhances problem-solving abilities [10], allows patients to interact with their HCPs without time or distance limitations [5], and may increase their willingness and engagement to adopt PD [11].

mHealth is a type of patient-centered care that regards patients and their families as core elements of the healthcare system through mobile and wireless devices [12], with a goal of motivating and encouraging patients to collaborate and engage in treatment plans [12,13,14,15,16,17]. mHealth promotes healthy behaviors, prevents acute and chronic diseases, facilitates self-management [18], and improves the quality of care and services [19,20]. It is a type of medical technology that utilizes mobile devices (e.g., smartphones, tablets, wearable biometric monitors/sensors, personal digital assistants, etc.) to motivate, remind, and increase self-management [21,22], adhere to treatment [23], practice healthy lifestyles [24], and improve quality of life [14,25,26]. Therefore, mHealth apps are gaining recognition and being widely used to promote the self-management of chronic illnesses, such as chronic obstructive pulmonary disease [27], chronic kidney disease [16], asthma [28], heart failure [29], and stress [30]. Most studies addressed the feasibility, usability, acceptability, and satisfaction of these apps [31,32]. The MARS has been widely used to assess the quality of mHealth apps [29,30].

Incorporating mHealth technology into the standard of care for patients still faces many challenges, including comprehensive evaluations of mHealth interventions that could motivate wider adoption of the technology. There have not been reviews of mHealth inventions and their assessments specific to the comprehensive self-management of PD, although some papers have discussed hemodialysis. The previous evaluation of mHealth apps was performed by a team of reviewers, mainly on the construction of the apps, instead of on the efficacy and benefits of using the apps in patient care and clinical settings [16]. Most app evaluation instruments are designed to assess feasibility and usability. However, there is still room for improvement in evaluating content across all aspects of care. Therefore, researchers conducted a literature review to integrate disease-related management domains for patients undergoing PD. Researchers have synthesized how patients should deal with PD in their daily lives. These comprehensive self-management domains included the following: (1) dialysis management: performing PD skills, recording the volume of fluid, preventing peritonitis, checking and tracking lab data, and managing symptoms [4,5,33]; (2) nutrition management: calculating and monitoring nutrition/dietary and food phosphorus content [4,13]; (3) exercise management: type, frequency, and amount of activity [34]; (4) medication management: taking medication regularly [13]; (5) physiological indicators: recording and/or tracing blood pressure and body weight, especially pre- and post-dialysis [34]; (6) laboratory values: checking and tracking lab data, such as creatinine and/or blood sugar levels [4,13]; (7) information: disease-related knowledge, estimated glomerular filtration rate (eGFR), and symptom management [4]; (8) interaction: communicating with HCPs, peers, or family [5,34,35]; and (9) other: medical appointments or social resources for achieving better health [13,36]. Patients improve their accuracy, adherence, self-care, and disease literacy through mHealth [13,37,38].

This study aimed to (1) search mobile apps for patients with PD on the Google Play app and iTunes App Store; (2) rate mobile apps according to engagement, functionality, aesthetics, information, subjective quality, and app specificity of the MARS APP Quality Scores; (3) investigate the domains targeted for the comprehensive self-management of PD in accordance with self-care guidelines; and (4) recommend suitable mobile applications to facilitate patient self-management and health.

## 2. Materials and Methods

### 2.1. Study Design

This study presents the perceived feasibility, usability, and effectiveness of applications designed for use by patients with PD. Researchers identified suitable apps downloaded from Google Play on Android and from the iTunes App Store on iOS, which, together, represent 99.23% of the smartphone app market share [39]. People are more willing to use smartphones for convenience, wide bandwidth, and fast internet speed to practice mHealth [40].

### 2.2. App Search Strategy

Considering that the accessibility, portability, flexibility, convenience, effectiveness, and usability of apps influence user willingness to use such technologies [18,29,40,41], the search, screening, assessment, and identification of domains adhering to self-management PD apps were conducted by a nursing professor (SM) and a researcher (ML) involved in identifying the applications related to patients with PD. Two reviewers downloaded and independently tested the criteria apps using an HTC U12 (Android 8.1) and iPhone 12 (iOS 14). From 3 to 16 June 2023, we conducted a thorough search of mobile apps. We included “peritoneal dialysis”, “dialysis”, “kidney disease”, and “chronic kidney disease” as search terms to search PD-related apps from patients’ perspectives in English and traditional Chinese on Android Google Play and the iOS iTunes App Store, which are the most accessible mobile app platforms. These search strings were obtained from several results discussed by the reviewers; the interactions with the mobile app platforms during the process were also investigated.

### 2.3. Selection Criteria

All identified apps were screened according to the titles, descriptions, target populations, screen snaps, and comments of the apps to determine whether they were appropriately used from the patients’ perspectives. The inclusion criteria were as follows: (1) development of PD patients; (2) provided in the traditional Chinese or English languages; (3) downloadable in the official Google Play or iTunes App Store; (4) downloadable by smartphone; and (5) free to download, as the cost of an app is a barrier to its use [10,42,43]. The exclusion criteria were as follows: (1) duplicate apps, including different versions and different names but the same content, appearing across platforms; (2) designed and used for HCPs; (3) cannot be downloaded; and (4) not written in English or traditional Chinese.

### 2.4. Data Extraction

The following app features from app stores and homepages were collected and recorded in Excel: app name, registration requirements, privacy policy, security, language, rating star, download times, version, last update, developer, and function from 3 to 16 June 2023. This study utilized the MARS instrument to evaluate the feasibility of apps and focused on comprehensive self-management for PD, which was derived from a literature review.

Two reviewers received standard training in MARS through YouTube instructional videos to understand how content and methodology were evaluated [44]. Two reviewers (SM and SF), specifically nursing professors, individually assessed the usability scores through MARS, coded the features, and investigated how many of the app contents matched the contents of self-management for PD from 1 August to 30 November 2023. Each reviewer initially assessed two randomly selected applications to assess inter-rater reliability. In cases of significant discrepancies among scores, discussions were conducted with a consensus approach to resolve scoring disparities; alternatively, a third reviewer was consulted to reach an agreement. For comprehensive integrated care in PD, the reviewers reached a consensus through discussion. The two reviewers then used content analysis to code for application contents. All data were coded in Microsoft Office Excel 2016. Cohen’s kappa coefficient was calculated to analyze the agreement between the MARS scores. A kappa coefficient greater than 0.8 represents an almost perfect agreement.

### 2.5. Measures of Rating Instrument

The MARS was developed by Stoyanov et al. (2015) and has been widely used to assess the design and usability of mHealth apps for different applications [44], such as gastrointestinal diseases [45], weight management [46], and chronic dialysis monitoring [20], to evaluate app quality. Correlations among the four objective, one subjective, one app-specific, and overall subscales were used to examine the validity. The Pearson correlation ranged from 0.643 to 0.800 and was shown to be significant. The test–retest reliability ranged from 0.78 to 1.000. A psychometric evaluation indicated that the MARS is a valid and reliable instrument for assessing app usability [47].

The MARS consists of four objectives (engagement, functionality, aesthetics, and information), one subjective quality, and one app-specific subscale. The four objective subscales comprise 19 items using a 5-point Likert scale ranging from 1 to 5 (1 = inadequate, 2 = poor, 3 = acceptable, 4 = good, and 5 = excellent). They include engagement (entertainment, interest, customization, interactivity, fit to the target group, personalization, interactivity, and target group), functionality (performance, ease of use, navigation, and gestural design), aesthetics (layout, graphics, and visual appeal), and information (accuracy of app description, goals, quality information, quantity information, visual information, credibility, and evidence-based information). The subjective quality subscale comprises three items (recommendation, likelihood of using the app in the next 12 months, payment, and star rating of the app). The app-specific subscale consists of six items with perceived impacts, including raising awareness, increasing knowledge, attitudes, intention to change, help-seeking behavior, and facilitating behavior change. A MARS score of over 3 points indicates acceptable quality [16,29]. Reviewers evaluated the apps for MARS ratings for at least 10 min.

### 2.6. Comprehensive Self-Management of PD

According to the relevant literature, researchers have compiled eight domains related to the self-management content of patients with PD. The eight domains include (1) dialysis management: performing PD skills, recording the volume of fluid, and preventing peritonitis; (2) nutrition management: calculating and monitoring nutrition/dietary status; (3) exercise management: type, frequency, and amount; (4) medication management: taking medication regularly; (5) physiological indicators: recording blood pressure and body weight, especially pre-dialysis and post-dialysis; (6) laboratory values: checking and tracking lab data; (7) information: disease-related knowledge, estimated glomerular filtration rate (eGFR), and symptom management; (8) interaction: communicating with HCP, peers, or family; and (9) other: medical appointments or social resources for achieving better health. Each app was scored on a scale from 1 to 8. The lowest score is one point, and the highest score is eight points.

### 2.7. Statistical Analysis

The characteristics, MARS scores, and overall comprehensive self-management of PD in the apps were collected in Excel. The above data were then imported into SPSS for descriptive statistics as well as to calculate the mean (standard deviation) of each subscale and the overall MARS score. The intra-class correlation coefficient (ICC) was analyzed to assess the inter-rater reliability among reviewers when rating the MARS. The overall comprehensive integrated care for PD is presented in terms of frequency and percentage.

## 3. Results

### 3.1. General Characteristics

By searching Google Play and the iTunes App Store using keywords, 828 potentially relevant applications were discovered. Eventually, 21 applications met the criteria for further investigation (Figure 1). The accuracy of the keyword searches on Apple iTunes was better than that on Android Google Play. Most applications on these two platforms do not provide user star ratings. Google Play includes information on download counts, whereas Apple iTunes does not. The majority of the applications identified involved kidney disease-related knowledge (10/21, 47.6%); followed by kidney function calculation (8/21; 38.1%); records (7/21; 33.3%); interaction (4/21; 19.0%); a checking, tracking, and/or reminders (3/21, 14.3%) function, as well as the phosphorus content of food (2/21, 9.5%) (Table 1).

### 3.2. Privacy and Security Features

The developers describe privacy policies and security on the app stores or websites. Of the 21 mHealth apps, 5 (23.8%) claim to have privacy protection, 11 (52.4%) collect or share data, and 5 (23.8%) are not clear. Twelve (57.1%) applications explain their security, eight (38.1%) apps do not provide explanations, and one (4.8%) cannot guarantee security (Table 1). Some applications allow users to decide whether to share data, such as the “Kidney Guide by Taipei Veterans General Hospital” app.

### 3.3. MARS App Quality Scores

The overall mean score was 3.58 (SD = 1.04) (Table 2), reaching an acceptable score of 3 on the MARS [16,29]; three apps scored below 3 points. These results are similar to those reported in other studies [22]. The highest score of the subscale is “function”, followed by “aesthetics”, while the lowest score was “subjective quality”. The highest-scoring applications were “Mizu-Your CKD companion” and “Kidney Guide by Taipei Veterans General Hospital”, while the lowest scores were for “Chronic Kidney Disease” and “Kidney Failure Risk Equation”. The ICC values ranged from 0.887 to 0.886, indicating good to excellent interrater reliability [48]. 

### 3.4. Comprehensive Self-Management of PD

Investigating the content of existing applications aimed at recommending self-management practices for patients with PD is crucial for improving patient care and ensuring that these apps effectively meet patients’ needs. In addition to being user-friendly, it is crucial for apps to encompass comprehensive PD care. In this study, disease-related information constituted the majority (18/21, 85.7%), followed by physiological indicators (7/21, 33.3%) and laboratory values (6/21, 28.6%). Medication management (2/21, 9.5%) and others (medical appointments or social resources) (2/21, 9.5%) are the least covered aspects (Table 3) (Figure 2).

## 4. Discussion

This study aimed to identify and discover suitable applications to recommend to patients with PD. The findings suggest that several applications, meeting the defined criteria for high quality, can be recommended to patients for use in English or traditional Chinese. These applications have received high ratings and provide rich content, making them suitable recommendations for patients. The research team developed specific search and evaluation strategies to discover high-quality applications that enable patients to access comprehensive care content. A quality assessment was conducted using the validated MARS scale, which assesses usability and acceptability based on measured engagement, functionality, aesthetics, information quality, subjective quality, and app specificity. Additionally, the comprehensive self-management approach for PD patients was utilized to examine the practicality of the applications within the context of disease-related management.

mHealth applications have been shown to be effective in CKD self-management. However, navigating through an increasing number of available applications to determine the most suitable one for patients can be challenging [39]. In the search process, the Apple App Store provides more accurate application choices than Google Play, indicating a need for enhanced precision in the latter [16,49].

MARS is an instrument used for assessing the quality of mobile health applications [38,50], primarily focusing on usability and accessibility. Among the identified applications, the overall average score on the MARS was higher than the recommended score by 3 points, which is consistent with the previous research on applications for CKD [16]. These applications scored highest in the functionality dimension, indicating a high level of performance in technology and design. However, scores in the engagement dimension, especially in the entertainment item, were lower; these applications primarily emphasize practicality with less emphasis on entertainment. There is scope for improvement in this entertainment aspect in the future. Among these apps, there were eight applications for estimated glomerular filtration rate (eGFR), which may have limited practicality for patients; they rely on hospital blood test results to calculate the eGFR value. This may have contributed to these applications ranking second to last on the app-specific subscale. Searching academic databases for mHealth reviews has proven to be useful [50]. However, apps obtained from app store platforms have not yet been studied, thus making it impossible to verify their effectiveness. This could result in them not being considered for high-quality evidence-based applications [38]. This may be because many applications are developed by private developers, thus making clinical research challenging [29].

Privacy and security significantly affect user willingness. Privacy refers to freedom from unauthorized intrusion and not sharing data with others. Security is a means of preventing the theft or hijacking of data or code within an application. Only five applications claim to prioritize privacy; the majority do not address it [38,50]. Some applications share user information with third parties, thereby raising concerns about ensuring privacy. In terms of security, more than half of the applications claim to have security features. However, there is a possibility of collecting device IDs or other ID-related information, as seen in apps like “eGFR Calculators Pro” (developed by iMedical Apps). This may bring potential security uncertainties [42,43].

Most of the applications cannot encompass all of the content related to peritoneal dialysis self-management. PD care should encompass residual kidney function, PD exchange volume/frequency and length, solution type, small solute clearance, nutritional status, lifestyle factors (e.g., exercise), medication, cardiovascular function (e.g., blood pressure), biochemical indicators, knowledge, and social support. This aligns with the content of the comprehensive self-management of PD in this study. Among the 21 reviewed applications, the most common content included estimating kidney function formula calculations and providing information about disease-related content. Some applications allow users to input records of dialysis fluid types, input and output volumes, physiological indicators, reminders or alerts regarding medications or symptom tracking, and interactions among patients, peers, and HCPs. Unfortunately, none of the reviewed apps can comprehensively include all of the mentioned content [4].

Although there was a total of 21 apps that met the search criteria, considering the MARS and the results of the self-management assessment for peritoneal dialysis integrity, the better-performing ones are “Mizu- Your CKD Companion” and “Kidney Guide by Taipei Veterans General Hospital”. Mizu stands out in terms of graphic quality, color coordination, and resolution, presenting a clear and concise overall style with good consistency. It is considered more interesting than other apps and includes a reminder notification feature that is expected to enhance patient self-management and adherence [39]. This application emphasizes that it was developed by an interdisciplinary team. Fan and Zhao (2022) [14] as well as Lukkanalikitkul et al. (2022) [33] indicated that applications developed by multidisciplinary teams are effective for optimizing both application and monitoring systems. The advantage of this approach is the integration of expertise and skills from different professional fields, which ensures that the content and functionality of the application are comprehensively considered. Additionally, applications developed by multidisciplinary teams often meet user needs better; experts from various fields can collaboratively design and evaluate an application, ensuring its alignment with clinical and user environments. “Kidney Guide” by Taipei Veterans General Hospital is a health-related app specifically developed and produced for PD patients. It includes features such as counseling, exercise tracking, and interactive functionalities. Moreover, it is compatible with existing healthcare systems [50]. Users have the flexibility to choose whether to connect their personal medical records, medications, and test results, which enables them to manage their health effectively. Real-time interactive communication with HCPs is a crucial application feature. Through data sharing, patients and HCPs can share patient health conditions, enabling problems to be addressed immediately and enhancing patient confidence as well as providing a sense of security [25]. This is a vital consideration for patients [11,29,38].

This study had some limitations. First, paying for downloads can be a barrier to using applications. Therefore, our search was limited to free apps. Whether paid applications differ from free applications in terms of functionality and self-management content should be discussed in the future. Second, the rapid development of applications in online stores is often dynamic, thus making systematic searches in app stores challenging. Third, some PD mHealth apps have been developed by universities or hospitals that have not entered the app market [45], so they cannot be accessed and assessed. Fourth, the reviewers of the application were people who only pretended to be patients rather than actual patients; therefore, their perspectives may differ from those of real patients in this study. This was beyond the scope of this study because researchers could not access mHealth apps that required payment, private subscriptions, and specific private practices used by hospitals.

## 5. Conclusions

Searching for specific terms or keywords in app stores may require browsing through hundreds of applications on Google Play and iTunes App Store, many of which remain irrelevant. The selection was limited to the information provided in the descriptions of each app available in the app store; only a few truly catered to the specific needs of patients with PD. An ideal mHealth app should be simple and intuitive to use. “Mizu- Your CKD Companion” and “Kidney Guide by Taipei Veterans General Hospital” are the most suitable apps for PD patients according to MARS APP Quality Scores and the comprehensive integrated care of PD. Progress in healthcare information technology has made healthcare applications increasingly streamlined and clear, thereby achieving ease of use. Patients with PD benefit from dedicated applications tailored to their specific needs [36]. Unfortunately, not all applications on app platforms have undergone scientific validation; therefore, it is questionable whether they truly have a beneficial impact on health. This issue may arise because most applications are constructed by private developers. In the future, patients should be included as members of application development teams to further refine and perfect their features.

## Figures and Tables

**Figure 1 healthcare-12-00719-f001:**
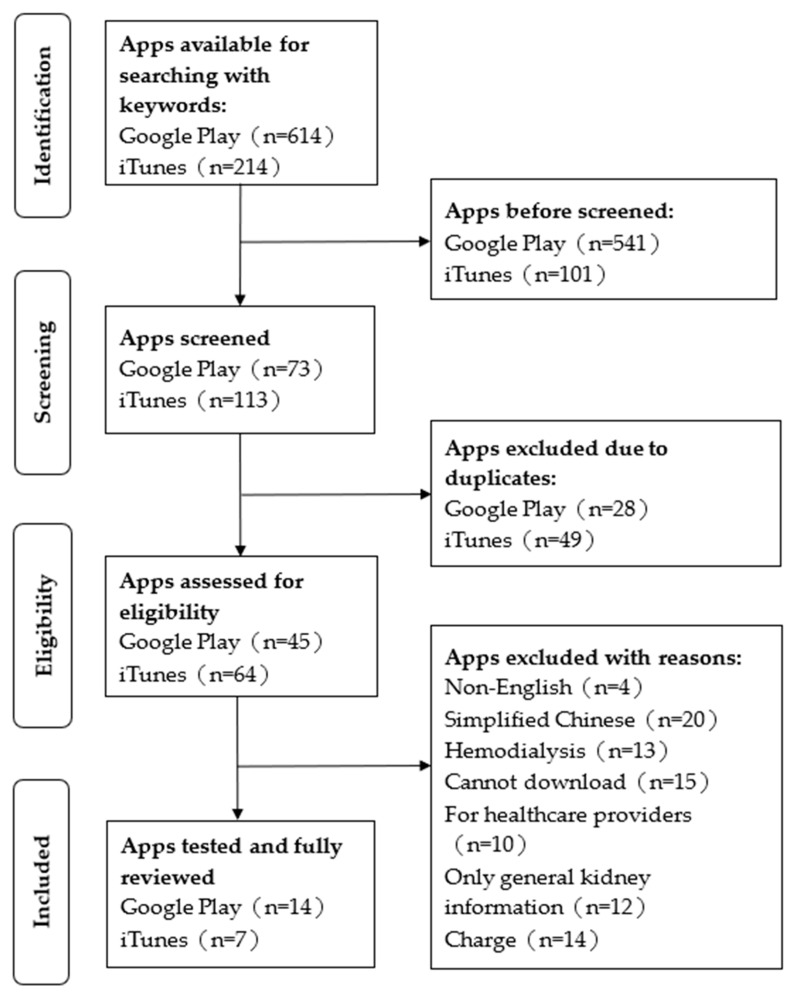
Screening process flowchart.

**Figure 2 healthcare-12-00719-f002:**
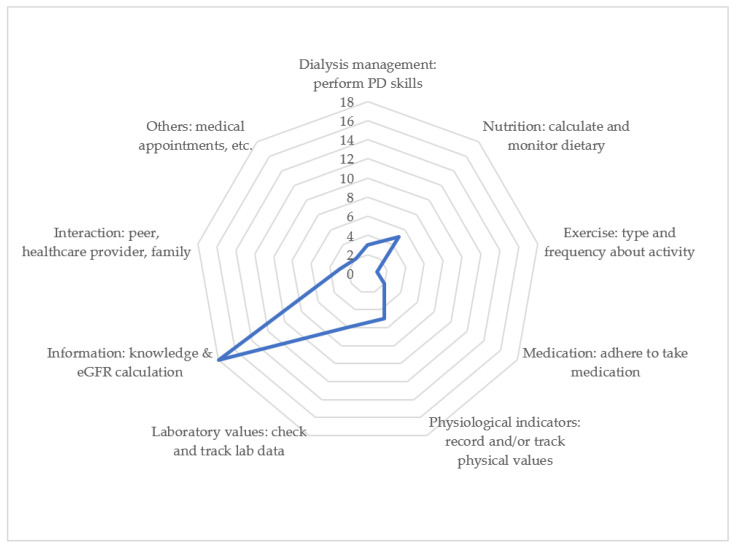
Radar chart of comprehensive integrated care of peritoneal dialysis among apps.

**Table 1 healthcare-12-00719-t001:** Mobile app characteristics.

App Name/Platform	Registration Requirements	Privacy	Security	Language	Rating Star	Download Times	Version	Last Update	Developer	Function
eGFR calculator /G & A	N/A	N/A	unknown	English	3.1	100,000+	N/A	10/11/2021	National Kidney Foundation	Knowledge Calculation
Dialysis calculator /G	N/A	N/A	unknown	English	4.4	10,000+	N/A	18/01/2021	Rodrigo Sepúlveda Palamara	Calculation
My Kidney friend /G	N/A	Yes	encryption	English	N/A	1000+	N/A	12/09/2022	Anastasiia Elci	Record
eGFR calculator pro /G	N/A	Shares information	encryption	English	5.0	100,000+	N/A	10/08/2021	iMedical Apps	Calculation
kidney graphs result for kidney	N/A	Shares information	encryption	English	N/A	50+	N/A	10/08/2021	Torches Inc.	Record
Kidney Guide by Taipei Veterans General Hospital /G	Identity card number + password	Shares information to NHIA (Taiwan)	encryption	Chinese	N/A	100+	N/A	27/07/2022	Taipei Veterans General Hospital	Knowledge Record Tracking Reminder Interaction
Mizu- Your CKD companion /G & A	email + password	Shares information	encryption	English	N/A	5000+	N/A	27/10/2023	Carealytix	Knowledge Record Tracking Reminder Interaction
eGFR Calculator (CKD-EPI) /G	N/A	Yes	unknown	English	N/A	10,000+	N/A	13/11/2021	MDApp+	Calculation
Kidney renal disease diet help /G	N/A	Shares ID	encryption	English	N/A	10,000+	N/A	21/06/2023	Data Recovery Software by RecoveryBull.com	Knowledge
RenalSense: Kidney Care App /G	N/A	Yes	encryption	English	N/A	50+	N/A	10/11/2023	Loop Systems	Knowledge Record
Chronic Kidney Disease /G	N/A	Yes	encryption	English	N/A	100+	N/A	08/04/2023	DevoDreamTeam	Knowledge
Kidney Failure Risk Equation /G	N/A	N/A	unknown	English	N/A	100+	N/A	23/08/2021	M. Parmar	Calculation
Kidney Care Community /G	N/A	Collects and shares information	encryption	English	N/A	100+	N/A	26/10/2021	Fresenius Medical Care North America	Knowledge Interaction
eGFR calculators pro /G	N/A	Collects information	encryption	English	N/A	100,000+	N/A	19/11/2021	iMedical Apps	Knowledge Calculation
PeriBuddy-managing peritonea /G & A	email + password	Collects information	encryption	English Chinese	N/A	N/A	0.9.16	3 years ago	Outsource ESD	Tracking Interaction (Share information with others)
Renal dialysis /A	email + password	Collects information	unknown	English	4.3	N/A	0.9.16	3 years ago	Coding Minds, Inc.	Record
Low Phosphorus Foods /A	N/A	N/A	unknown	English	N/A	N/A	1.3.1	3 years ago	Nasir Hussain	Record Check the phosphorus content of food
Our Journey with PD /A	N/A	N/A	unknown	English	N/A	N/A	2.1	3 years ago	Phoenix Children’s Hospital, Inc.	Knowledge
Kidney Diet Friendly Recipes /G & A	N/A	Collects information	Not guarantee	English	4.4	1000+	3.1.0	1 week ago	Prestige Worldwide Apps LLC	Knowledge Check the phosphorus content of food
eGFR calculators pro /A	N/A	Traces and collects information	unknown	English	N/A	N/A	3.3	2 years ago	Putu Angga Risky Raharja	Calculation
GFR Easycalc /G & A	N/A	Yes	encryption	English	N/A	10+	N/A	N/A	Louis Janssens	Calculation

*Notes:* G: Google Play; A: iTunes App Store; N/A: Not applicable; NHIA: National Health Insurance Administration.

**Table 2 healthcare-12-00719-t002:** The Mobile App Rating Scale quality rating.

App Name	Engagement	Functionality	Aesthetics	Information	Subjective Quality	App Specificity	Overall
	Mean	Mean	Mean	Mean	Mean	Mean	Mean
eGFR calculators	3.20	5.00	4.50	4.19	4.50	3.83	4.20
Dialysis calculator	3.20	5.00	4.50	4.20	4.50	3.83	4.20
My Kidney friend	2.80	5.00	4.50	3.89	4.13	3.33	3.94
eGFR calculator pro	2.80	5.00	4.50	3.78	2.75	4.08	3.82
kidney graphy result for kidney	2.90	5.00	4.50	3.83	4.13	3.00	3.89
Kidney Guide by Taipei Veterans General Hospital	3.60	5.00	5.00	4.67	5.00	4.92	4.70
Mizu- Your CKD companion	4.00	5.00	5.00	4.76	5.00	5.00	4.79
eGFR Calculator (CKD-EPI)	2.80	5.00	4.33	3.52	3.25	2.50	3.57
kidney renal disease diet help	2.80	4.00	4.33	2.99	1.25	3.00	3.06
Renal Sense: Kidney Care App	2.80	4.00	4.33	2.83	1.25	2.50	2.95
Chronic Kidney Disease	2.40	3.75	3.00	2.28	1.00	1.50	2.32
Kidney Failure Risk Equation	2.40	3.75	3.00	2.34	1.00	1.50	2.33
Kidney Care Community	2.80	4.50	4.00	3.78	3.13	4.33	3.76
eGFR calculator pro	2.80	5.00	3.83	3.68	3.75	3.00	3.68
Peri Buddy-managing peritonea	3.00	5.00	4.33	3.77	3.63	3.00	3.79
Renal dialysis	3.10	5.00	4.33	4.04	4.25	4.00	4.12
Low Phosphorus Foods	3.50	5.00	4.33	4.31	3.75	5.00	4.31
Our Journey with PD	3.00	4.50	4.83	4.17	4.00	4.83	4.22
Kidney Diet Friendly Recipes	2.30	4.50	4.00	3.41	2.75	3.08	3.34
eGFR calculator pro	2.80	5.00	4.50	3.78	2.75	4.08	3.82
GFR Easycalc	2.80	5.00	4.50	3.78	2.75	4.08	3.82
Mean (SD)	2.94 (0.40)	4.71 (0.46)	4.29 (0.52)	3.71 (0.65)	3.26 (1.27)	3.54 (1.04)	3.74 (0.65)

**Table 3 healthcare-12-00719-t003:** The domains of comprehensive self-management of PD included in the study.

App Name	Dialysis Management: Perform PD Skills, Record the Volume of Fluid, Prevent Peritonitis	Nutrition Management: Calculate and Monitor Nutrition/Diet and Food Phosphorus Content	Exercise Management: Type, Frequency, and Amount of Activity	Medication Management: Taking Medication Regularly	Physiological Indicators: Record and/or Trace Blood Pressure and Body Weight, Especially Pre-Dialysis and Post-Dialysis	Laboratory Values: Check and Track Lab Data, Such as Creatinine and/or Blood Sugar Levels	Information: Knowledge and eGFR Calculation	Interaction: Peer, HCP, Family	Others: Medical Appointments, Social Resources, Quality of Life	Total Score
eGFR calculators							✓			1
Dialysis calculator	✓	✓			✓	✓	✓			5
My Kidney friend					✓	✓				2
eGFR calculators pro							✓			1
kidney graphy result for kidney						✓	✓			2
Kidney Guide by Taipei Veterans General Hospital		✓	✓	✓	✓	✓	✓	✓	✓	8
Mizu- Your CKD companion		✓		✓	✓	✓	✓	✓	✓	7
eGFR Calculator (CKD-EPI)							✓			1
kidney renal disease diet help							✓			1
RenalSense: Kidney Care App					✓		✓			2
Chronic Kidney Disease							✓			1
Kidney Failure Risk Equation							✓			1
Kidney Care Community							✓	✓		2
eGFR calculators pro							✓			1
PeriBuddy-managing peritonea	✓				✓		✓			3
Renal dialysis	✓				✓	✓				3
Low Phosphorus Foods		✓					✓			2
Our Journey with PD							✓			1
Kidney Diet Friendly Recipes	✓	✓								2
eGFR calculators pro							✓			1
GFR Easycalc							✓			1
Total score	4	5	1	2	7	6	18	3	2	48

## Data Availability

Data will be made available upon request.

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
