# Peer review of "Mobile Apps for Patients with Peritoneal Dialysis: Systematic App Search and Evaluation"

_healthcare, 2024, doi:10.3390/healthcare12070719_

Round 1

Reviewer 1 Report

Comments and Suggestions for Authors

In order to support PD patients’ self-care and health, the paper made a review of the freely available and suitable mobile applications (from Google Play on Android and from iTunes App Store on iOS) for patients with PD and rated a number of selected apps (considering the patients’ needs related functionality, information provision, quality and specificity). They explored the feasibility, usability, and effectiveness of these apps for use by patients with PD and offered recommendations for apps that would be useful to improve PD patients’ health.

ABSTRACT

1.     The abstract is too long. As stated in Healthcare journal's "Instructions for Authors”, “The abstract should be a total of about 200 words maximum” (https://www.mdpi.com/journal/healthcare/instructions). In its current form, there are more than 300 words;

2.     It needs to be restructured in order to enhance its clarity of the presentation;

3.     It is necessary to specify the period of performing the review of freely available mobile applications for patients with PD.

INTRODUCTION:

1.     The authors make a thorough review/evaluation of the state of the research on mobile applications for PD of the most important recent articles;

2.     It is necessary to specify the main questions addressed by the research presented in this paper that is its objectives/aims of the paper. This is also the request specified in Healthcare journal's "Instructions for Authors” (Introduction “should define the purpose of the work and its significance, including specific hypotheses being tested” (https://www.mdpi.com/journal/healthcare/instructions));

3.     The objectives of the paper must to be analyzed because information concerning the aims of the paper is included in different parts of the text (lines 17-20, 96-100,  225-227, 239), but with a slightly different content. Please review/rewrite them carefully;

4.     Please specify why you consider that your research is original and relevant to the field and is added to the subject area compared with other cited materials.

MATERIALS AND METHODS

1.     The methodological strategy is explained in detail;

2.     I believe it would be better to incorporate the content of lines 96–100, which comprise the first paragraph of the chapter "2. Materials and Methods", with a restructured format (see comment 3 of the Introduction chapter), in the Introduction chapter;

3.     The clarity of the presentation would be improved if diagrams are included into the presentation;

4.     When presenting “The apps search strategy”, the existence of “a panel of experts, including nurses and chronic kidney disease specialists” (lines 114-115) is specified. It is necessary to provide more information concerning the composition of this panel as: number of person from each category, method/criteria of selection, period of activity.

RESULTS

1.     The findings and the authors' interpretation are succinctly and precisely described;

2.     The authors specify that “experts from various fields can collaboratively design and evaluate an application, ensuring its alignment with clinical and user environments” (lines 3045-305). It is necessary to specify the number of experts and which are the selection criteria.

DISCUSSION

1.     A summary of the specific methods and results is necessary to be included at the beginning of “Discussion” chapter;

2.     The “Strengths and limitations” of the work have to be included in the “Discussion” Chapter instead of “Conclusion” chapter;

3.     The authors are requested to specify if the existing limitations lead to future work and, if so, describe it shortly.

CONCLUSION

1.     Please specify if all the aims were addressed.

REFERENCES

1.     References are up-to-date and appropriate to the subject;

2.     Please adhere to the Reference format guidelines provided in the Manuscript Preparation section (https://www.mdpi.com/journal/healthcare/instructions).

ABBREVIATIONS

1.     The abbreviation Mobile App Rating Scale (MARS) is used once in the Abstract (line 22) and twice in the chapters of the paper (lines 153 and 292). In line 292, please write only MARS because the abbreviation is already defined.

I hope my feedback is useful to the authors in improving their paper and wish them all the best in pursuing this important area of research.

Author Response

ABSTRACT

1. The abstract is too long. ….. In its current form, there are more than 300 words;

The abstract has been revised.

Page 1

2. It needs to be restructured in order to enhance its clarity of the presentation;

The abstract has been revised.

Page 1

3. It is necessary to specify the period of performing the review of freely available mobile applications for patients with PD.

Added period.

Page 3

4. It is necessary to specify the main questions addressed by the research presented in this paper that is its objectives/aims of the paper. This is also the request specified in Healthcare journal's "Instructions for Authors” (Introduction “should define the purpose of the work and its significance, including specific hypotheses being tested” (https://www.mdpi.com/journal/healthcare/instructions));

This part has been revised.

Page 2

5. The objectives of the paper must to be analyzed because information concerning the aims of the paper is included in different parts of the text (lines 17-20, 96-100, 225-227, 239), but with a slightly different content. Please review/rewrite them carefully;

This part has been revised.

Page 2

6. Please specify why you consider that your research is original and relevant to the field and is added to the subject area compared with other cited materials.

This part has been revised.

Page 2-3

MATERIALS AND METHODS

1. The methodological strategy is explained in detail;

Thanks!

2. I believe it would be better to incorporate the content of lines 96–100, which comprise the first paragraph of the chapter " 2. Materials and Methods", with a restructured format (see comment 3 of the Introduction chapter), in the Introduction chapter;

This part has been revised.

Page 3

3. The clarity of the presentation would be improved if diagrams are included into the presentation;

Figure 1 Revised

Page 6-Figure 1

4. When presenting “The apps search strategy”, the existence of “a panel of experts, including nurses and chronic kidney disease specialists” (lines 114-115) is specified. It is necessary to provide more information concerning the composition of this panel as: number of persons from each category, method/criteria of selection, period of activity.

This part has been revised.

Page 3

RESULTS

1. The findings and the authors' interpretation are succinctly and precisely described;

Thanks!

2. The authors specify that “experts from various fields can collaboratively design and evaluate an application, ensuring its alignment with clinical and user environments” (lines 304-305). It is necessary to specify the number of experts and which are the selection criteria.

This part has been revised.

Page 13

DISCUSSION

1. A summary of the specific methods and results is necessary to be included at the beginning of “Discussion” chapter;

This part has been revised.

Page 12-13

2. The “Strengths and limitations” of the work have to be included in the “Discussion” Chapter instead of “Conclusion” chapter;

This part has been revised.

Page 12-13

3. The authors are requested to specify if the existing limitations lead to future work and, if so, describe it shortly.

This part has been revised.

Page 13

CONCLUSION

1. Please specify if all the aims were addressed.

This part has been revised.

Page 13

REFERENCES

1. References are up-to-date and appropriate to the subject;

Thanks!

Page 16

2. Please adhere to the Reference format guidelines provided in the Manuscript Preparation section (https://www.mdpi.com/journal/healthcare/instructions).

ABBREVIATIONS

1. The abbreviation Mobile App Rating Scale (MARS) is used once in the Abstract (line 22) and twice in the chapters of the paper (lines 153 and 292). In line 292, please write only MARS because the abbreviation is already defined.

I hope my feedback is useful to the authors in improving their paper and wish them all the best in pursuing this important area of research.

This part has been revised.

Page 4-5

Reviewer 2 Report

Comments and Suggestions for Authors

Thank you for the opportunity to review this interesting manuscript. I would like to give some suggestions.

TITLE

I think the title should be better written. Please pay attention to the definition of “content analysis”.

MANUSCRIPT

-The objectives must be reviewed. Please pay attention to the definition of “to explore”.

Lines 17-20: “Objective: Reviewing comprehensive, free mobile applications for patients with PD on the market. Recommending suitable mobile applications to facilitate patient self-care and health. Additionally, explore and determine whether there is existing research evidence supporting the improvement of health outcomes through the use of these applications”.

Lines 96-101: “The aims of this study were to: (1) explore a comprehensive review of mobile apps for patients with PD on the Google Play and iTunes App Store; (2) rate mobile apps according to engagement, functionality, aesthetics, information, subjective quality, and app specificity; (3) investigate the domains targeted for comprehensive integrated care of PD in accordance with self-care guidelines; and (4) explore and identify appropriate evidence- based apps to improve health.”

Lines 225-227: “Therefore, this study aims to investigate the content of existing apps that satisfy patient needs. In addition to being user-friendly, it is crucial for apps to encompass comprehensive PD care”

Line 239: “This study aimed to identify and discover suitable applications for patients with PD.”

-Please review MARS and references:

Lines 153-156: “The Mobile App Rating Scale (MARS) was developed by Stoyanov et al. (2015) and has been widely used to assess the design and usability of mHealth apps [46], such as gastrointestinal diseases [47], weight management (Bardus et al., 2016)[46], and chronic  dialysis monitoring [20] to evaluate app quality.”

Lines 242-245:The MARS scale assesses the usability and acceptability of applications based on measured engagement, functionality, aesthetics, information quality, subjective quality, and app specificity.”

Lines 253-254:MARS is an instrument used for assessing the quality of mobile health applications [39,50], primarily focusing on usability and accessibility.”

ABSTRACT

-Please, enter the date of the study.

- Lines 17-18: “… free mobile applications for patients with PD on the market.”  I think the words "on the market" are so vague. The authors could describe it better.

INTRODUCTION

-After the description of what remains to be known, the authors could present the objectives of the study.

-Line 53: “… mHealth continually monitors…” Please write here “Mobile health (mHealth)” and not in line 57: “Mobile health (mHealth) is a type of …”

MATERIAL AND METHODS

-Please, enter the date of the different steps of the study.

-Lines 96-101: “The aims of this study were to: (1) explore a comprehensive review of mobile apps for patients with PD on the Google Play and iTunes App Store; (2) rate mobile apps according to engagement, functionality, aesthetics, information, subjective quality, and app specificity; (3) investigate the domains targeted for comprehensive integrated care of PD in accordance with self-care guidelines; and (4) explore and identify appropriate evidence- based apps to improve health.”    The objectives may be described in the "Introduction" section.

-Lines 103-104: “This study explored the perceived feasibility, usability, and effectiveness of applications designed for use by patients with PD.”  Did this study explore or identify?

-Lines 110-125: The text is confusing. Please rewrite it

-Lines 112-117:

Two reviewers identified the applications related to PD patients. Who were the reviewers?

“The search, screening, assessment, and identification of domains adhering to self-care PD apps were conducted by a panel of experts, including nurses and chronic kidney disease specialists. How were the experts recruited? How many experts participated in the study?

“Two reviewers downloaded the required apps, tested the usability, coded the features, and investigated how many of the app contents matched the contents of self-care PD.” Who were the reviewers?

RESULTS

-The authors did not describe the sociodemographic data of the experts involved in the study, allowing to prove experience and strong contribution to the study.

-Table 1. Please describe “NIH”

DISCUSSION

-The 1st paragraph could include the summary of the results.

-Line 292: “…Mobile App Rating Scale (MARS)…”    The authors had already written “Mobile App Rating Scale (MARS)”, so they didn't need to do it again. Please write MARS only.

-“Strengths and limitations” could be described in "Discussion" section. Another limitation may have been the recruitment of experts in the study.

CONCLUSION

The authors could briefly describe the results of their study.

It was not supposed to describe references in the conclusions

Please rewrite this section.

Author Response

Reviewer Comment

Author Response

Page number

Reviewer #3

TITLE

I think the title should be better written. Please pay attention to the definition of “content analysis”.

This part has been revised.

Page 1

MANUSCRIPT

-The objectives must be reviewed. Please pay attention to the definition of “to explore”.

Lines 17-20: “Objective: Reviewing comprehensive, free mobile applications for patients with PD on the market. Recommending suitable mobile applications to facilitate patient self-care and health. Additionally, explore and determine whether there is existing research evidence supporting the improvement of health outcomes through the use of these applications”.

This part has been revised.

Page 2

Lines 96-101: “The aims of this study were to: (1) explore a comprehensive review of mobile apps for patients with PD on the Google Play and iTunes App Store; (2) rate mobile apps according to engagement, functionality, aesthetics, information, subjective quality, and app specificity; (3) investigate the domains targeted for comprehensive integrated care of PD in accordance with self-care guidelines; and (4) explore and identify appropriate evidence- based apps to improve health.”

This part has been revised.

Page 3

Lines 225-227: “Therefore, this study aims to investigate the content of existing apps that satisfy patient needs. In addition to being user-friendly, it is crucial for apps to encompass comprehensive PD care”

This part has been revised.

Page 3

Line 239: “This study aimed to identify and discover suitable applications for patients with PD.”

-Please review MARS and references:

This part has been revised.

Page 12

Lines 153-156: “The Mobile App Rating Scale (MARS) was developed by Stoyanov et al. (2015) and has been widely used to assess the design and usability of mHealth apps [46], such as gastrointestinal diseases [47], weight management (Bardus et al., 2016) [46], and chronic dialysis monitoring [20] to evaluate app quality.”

This part has been revised.

Page 4

Lines 242-245: “The MARS scale assesses the usability and acceptability of applications based on measured engagement, functionality, aesthetics, information quality, subjective quality, and app specificity.”

This part has been revised.

Page 12

Lines 253-254: “MARS is an instrument used for assessing the quality of mobile health applications [39,50], primarily focusing on usability and accessibility.”

This part has been revised.

Page 12

ABSTRACT

-Please, enter the date of the study.

- Lines 17-18: “… free mobile applications for patients with PD on the market.”  I think the words "on the market" are so vague. The authors could describe it better.

Added the date.

Revised.

Page 1

INTRODUCTION

-After the description of what remains to be known, the authors could present the objectives of the study.

This part has been revised.

Page 1-2

-Line 53: “… mHealth continually monitors…” Please write here “Mobile health (mHealth)” and not in line 57: “Mobile health (mHealth) is a type of …”

This part has been revised.

Page 2

MATERIAL AND METHODS

-Please, enter the date of the different steps of the study.

Added.

Page 2

-Lines 96-101: “The aims of this study were to: (1) explore a comprehensive review of mobile apps for patients with PD on the Google Play and iTunes App Store; (2) rate mobile apps according to engagement, functionality, aesthetics, information, subjective quality, and app specificity; (3) investigate the domains targeted for comprehensive integrated care of PD in accordance with self-care guidelines; and (4) explore and identify appropriate evidence- based apps to improve health.” The objectives may be described in the "Introduction" section.

This part has been revised.

Page 3

-Lines 103-104: “This study explored the perceived feasibility, usability, and effectiveness of applications designed for use by patients with PD.”  Did this study explore or identify?

This part has been revised.

Page 3

-Lines 110-125: The text is confusing. Please rewrite it

This part has been revised.

Page 3

-Lines 112-117:

“Two reviewers identified the applications related to PD patients. Who were the reviewers?

“The search, screening, assessment, and identification of domains adhering to self-care PD apps were conducted by a panel of experts, including nurses and chronic kidney disease specialists.” How were the experts recruited? How many experts participated in the study?

“Two reviewers downloaded the required apps, tested the usability, coded the features, and investigated how many of the app contents matched the contents of self-care PD.” Who were the reviewers?

This part has been revised.

Page 3

RESULTS

-The authors did not describe the sociodemographic data of the experts involved in the study, allowing to prove experience and strong contribution to the study.

Added.

Page 4

-Table 1. Please describe “NIH”

This part has been revised.

Table 1.

DISCUSSION

-The 1st paragraph could include the summary of the results.

This part has been revised.

Page 4

-Line 292: “…Mobile App Rating Scale (MARS)…”    The authors had already written “Mobile App Rating Scale (MARS)”, so they didn't need to do it again. Please write MARS only.

This part has been revised.

Page 13

“Strengths and limitations” could be described in "Discussion" section. Another limitation may have been the recruitment of experts in the study.

This part has been revised.

Page 12-13

CONCLUSION

The authors could briefly describe the results of their study.

It was not supposed to describe references in the conclusions

Please rewrite this section.

This part has been revised.

Page 13

Reviewer 3 Report

Comments and Suggestions for Authors

This paper assesses mobile apps for PD management in terms of their engagement, functionality, aesthetics, information quality, and alignment with integrated care standards. The paper is concise and short in terms of written content, but it provides an engaging discussion. Here are some recommendations for improvement.

- line 57-70: Authors can include the following article to their list:

Connelly K, Siek KA, Chaudry B, Jones J, Astroth K, Welch JL. An offline mobile nutrition monitoring intervention for varying-literacy patients receiving hemodialysis: a pilot study examining usage and usability. Journal of the American Medical Informatics Association. 2012 Sep 1;19(5):705-12.

- line 164 - 169: authors needs to clarify where the specific aspects for various objectives are derived from. For example, the aspects for engagement are entertainment, interest, customization, interactivity, fit to the target group, personalization, interactivity, and target group. Are these aspects based on evidence?

- lines 177-189: References are needed to support the self-management explanation.

- The caption for Figure 2 is currently unclear and should be rewritten.

- The last aim of the study stated on line 100-101, i.e.  identifying evidence-based apps that can improve PD patient health outcomes has not been fulfilled. How do authors expect to answer this question without patient involvement or feedback.

Author Response

Reviewer Comment

Author Response

Page number

Reviewer #4

- line 57-70: Authors can include the following article to their list: Connelly K, Siek KA, Chaudry B, Jones J, Astroth K, Welch JL. An offline mobile nutrition monitoring intervention for varying-literacy patients receiving hemodialysis: a pilot study examining usage and usability. Journal of the American Medical Informatics Association. 2012 Sep 1;19(5):705-12.

This part has been revised.

Page 2

- line 164 - 169: authors needs to clarify where the specific aspects for various objectives are derived from. For example, the aspects for engagement are entertainment, interest, customization, interactivity, fit to the target group, personalization, interactivity, and target group. Are these aspects based on evidence?

The MARS was developed by Stoyanov et al. (2015) and has been widely used to assess the design and usability of mHealth apps. A psychometric evaluation indicated that the MARS is a valid and reliable instrument for assessing app usability [reference 48].

Page 4

- lines 177-189: References are needed to support the self-management explanation.

This part has been revised.

Page 4

- The caption for Figure 2 is currently unclear and should be rewritten.

This part has been revised.

Figure 2

- The last aim of the study stated on line 100-101, i.e.  identifying evidence-based apps that can improve PD patient health outcomes has not been fulfilled. How do authors expect to answer this question without patient involvement or feedback.

This part has been revised.

Page 3

Line 51: "These factors affect the adoption of HD rather than PD." It is not clear what ‘these factors’ relate to.

This part has been revised.

Page 2

Line 53: I think you mean reduced medical expenses? Please clarify.

This part has been revised.

Page 2

Line 77: Please consider replacing instead of. This is blaming the other authors which if they did not intend to look at efficacy, then they cannot be blamed for this. What you can say is something like: however the efficacy and benefits of using apps …has not been investigated – something along those lines.

Revised. Very useful suggestion. Thanks!

Page 2

Line 130: Do you mean PD for patients?

Revised.

Page 3

Comments on the Quality of English Language

Good quality of English, though perhaps further proof-reading, especially in the first sections, might be of us.

Thank you for your suggestions, which help make our article clearer.

Reviewer 4 Report

Comments and Suggestions for Authors

Thank you for your paper and I enjoyed reading it. I wish you every success as you continue working in this area.

Overall, this paper reflects a systematic approach to gathering data. There are some areas that would (I think) benefit from further clarity, and I have made some suggestions below.

The background set the scene, and there could be a little more clarity around what other authors did or did not do, and that would help articulate the gap that your review would like to fill.

The materials and methods section included a good level of detail. There was a lack of clarity in some sentences (see below). The search terms were relevant and appropriate as were the inclusion/exclusion criteria.

Lines 112 and 115: You refer to two reviewers. Did these reviewers work independently, or together?

Line 143: How many reviewers were there?

Line 146: Can you please clarify what you mean by scores as there is no grading criteria presented. Do you mean the MARS rating scale? If so, state this for clarity. Also, can you please clarify how you defined a significant discrepancy versus a non-significant discrepancy.

The description of how the MARS tool was used, was clear.

How did you assess the subjective subscale? You mentioned checking reliability ahead of time, which was great. Can you please include a little detail on how this was maintained during the data extraction/ data collection/ grading phase? To ensure accuracy, it would be normal to have about 10% of the dataset cross-checked by another researcher, or have a number of the apps blindly assessed by more than one reviewer. Can you please confirm if you did this (and how), and if not, then it could perhaps be stated as a limitation of the study.

You also said that you created the CIC for PD from the literature. Has this been published anywhere? Have you tested this in any way? Were the criteria weighted for importance?

Line 174: Are the evaluators the same people as the reviewers?

Section 2.6. Some or all of this seems to be a repeat of lines 81 – 92. I suggest only including this information once.

Line 210: Of the 11 apps that collected or shared data, was the user offered any choice in this? Could users decline allowing the app to share data?

Figure 1 has an error. Apps assess = 45 + 64 = 109. Apps excluded = 74. 109 - 74 = 35, but you referred to 21 which matches the 14 +7 in the apps tested box. Somewhere there are 14 missing items.

Table 1 is useful and comprehensive. I wonder if it would be useful to use the same terms in the function column, as you use in the CIC of PD. For example, you state calculation, but this could refer to either Dialysis Management or Nutrition Management. Record seems to refer to physiological indicators and laboratory values. I think there could be more consistency between the terms used in the text analysis and tables.

Table 3 is also useful, and it may be useful to group similar characteristics together, perhaps aligned along the same lines as the CIC of PD, so all the calculator features together etc.

Discussion:

Lines 259 – 261: I am not sure why there is this focus on entertainment, especially as the need for entertainment was not mentioned previously.

Line 289: This concluding line in the paragraph seems to contradict the first sentence (line 279) when you state that there is a slight deficiency.

Conclusions: The first two sentences read like a limitations section. I would also move this paragraph to the end, and move the strengths and limitations section up.

In the discussion, I would have expected a little more detail on what future apps should include. Looking at Figure 2 and Table 3, there are a number of issues that you thought should be there, but are not, so further discussion on these would be useful.

Minor editing

Line 50: "Additionally, peritonitis may occur when patients perform 50 poorly [7]." I would re-phrase to avoid blaming the patient so directly. Perhaps include something like when a patient’s technique needs to be improved – or something along those lines.

Line 51: "These factors affect the adoption of HD rather than PD." It is not clear what ‘these factors’ relate to.

Line 53: I think you mean reduced medical expenses? Please clarify.

Line 77: Please consider replacing instead of. This is blaming the other authors which if they did not intend to look at efficacy, then they cannot be blamed for this. What you can say is something like: however the efficacy and benefits of using apps …has not been investigated – something along those lines.

Line 130: Do you mean PD for patients?

Comments on the Quality of English Language

Good quality of English, though perhaps furtehr proof-reading, especially in the first sections, might be of us.

Author Response

Reviewer Comment

Author Response

Page number

Reviewer #5

The materials and methods section included a good level of detail. There was a lack of clarity in some sentences (see below). The search terms were relevant and appropriate as were the inclusion/exclusion criteria.

Lines 112 and 115: You refer to two reviewers. Did these reviewers work independently, or together?

This part has been revised.

Page 3

Line 143: How many reviewers were there?

This part has been revised.

Page 4

Line 146: Can you please clarify what you mean by scores as there is no grading criteria presented. Do you mean the MARS rating scale? If so, state this for clarity. Also, can you please clarify how you defined a significant discrepancy versus a non-significant discrepancy.

This part has been revised.

Page 4

The description of how the MARS tool was used, was clear.

How did you assess the subjective subscale? You mentioned checking reliability ahead of time, which was great. Can you please include a little detail on how this was maintained during the data extraction/ data collection/ grading phase? To ensure accuracy, it would be normal to have about 10% of the dataset cross-checked by another researcher, or have a number of the apps blindly assessed by more than one reviewer. Can you please confirm if you did this (and how), and if not, then it could perhaps be stated as a limitation of the study.

You also said that you created the CIC for PD from the literature. Has this been published anywhere? Have you tested this in any way? Were the criteria weighted for importance?

Cohen's kappa coefficient was calculated to analyze the agreement between the MARS scores.

No. The researchers attempted to obtain information on peritoneal dialysis care from the International Society for Peritoneal Dialysis (website: https://ispd.org/). Unfortunately, it appears that the website is no longer accessible. Therefore, the content on peritoneal dialysis care was synthesized from relevant literature instead.

Page 4-5

Line 174: Are the evaluators the same people as the reviewers?

Section 2.6. Some or all of this seems to be a repeat of lines 81 – 92. I suggest only including this information once.

This part has been revised.

Page 4

Line 210: Of the 11 apps that collected or shared data, was the user offered any choice in this? Could users decline allowing the app to share data?

This part has been revised.

Page 5

Figure 1 has an error. Apps assess = 45 + 64 = 109. Apps excluded = 74. 109 – 74 = 35, but you referred to 21 which matches the 14 +7 in the apps tested box. Somewhere there are 14 missing items.

The total number is incorrect due to the truncation of the bottom right corner. Thank you for pointing that out. It has been corrected.

Figure 1

Table 1 is useful and comprehensive. I wonder if it would be useful to use the same terms in the function column, as you use in the CIC of PD. For example, you state calculation, but this could refer to either Dialysis Management or Nutrition Management. Record seems to refer to physiological indicators and laboratory values. I think there could be more consistency between the terms used in the text analysis and tables.

The functions of MARS and CIC differ somewhat. MARS focuses more on wireless transmission of electronic records, while CIC solely discusses PD disease management and healthcare.

Table 1

Table 3 is also useful, and it may be useful to group similar characteristics together, perhaps aligned along the same lines as the CIC of PD, so all the calculator features together etc.

MARS emphasizes electronic network healthcare functions, hence the emphasis on reminders, recording, tracking, etc. CIC focuses on disease management and healthcare. Therefore, there are still some differences between the two.

Table 3

Discussion:

Lines 259 – 261: I am not sure why there is this focus on entertainment, especially as the need for entertainment was not mentioned previously.

On commercial platforms, applications with entertainment value are more likely to attract consumers or prompt them to make purchases. However, if an application is developed by a hospital or academic institution, it may appear less entertaining to most users. Nevertheless, patients may still be willing to use it at the request of healthcare professionals or researchers.

Page 12

Line 289: This concluding line in the paragraph seems to contradict the first sentence (line 279) when you state that there is a slight deficiency.

This part has been revised.

Page 12-13

Conclusions

The first two sentences read like a limitations section. I would also move this paragraph to the end, and move the strengths and limitations section up.

This part has been revised.

Page 13

In the discussion, I would have expected a little more detail on what future apps should include. Looking at Figure 2 and Table 3, there are a number of issues that you thought should be there, but are not, so further discussion on these would be useful.

This part has been revised.

Table 2 and Table 3

Minor editing

Line 50: "Additionally, peritonitis may occur when patients perform 50 poorly [7]." I would re-phrase to avoid blaming the patient so directly. Perhaps include something like when a patient’s technique needs to be improved – or something along those lines.

This part has been revised.

Page 2

Line 51: "These factors affect the adoption of HD rather than PD." It is not clear what ‘these factors’ relate to.

This part has been revised.

Page 2

Line 53: I think you mean reduced medical expenses? Please clarify.

This part has been revised.

Page 2

Line 77: Please consider replacing instead of. This is blaming the other authors which if they did not intend to look at efficacy, then they cannot be blamed for this. What you can say is something like: however the efficacy and benefits of using apps …has not been investigated – something along those lines.

Revised. Very useful suggestion. Thanks!

Page 2

Line 130: Do you mean PD for patients?

Revised.

Page 3

Comments on the Quality of English Language

Good quality of English, though perhaps further proof-reading, especially in the first sections, might be of us.

Thank you for your suggestions, which help make our article clearer.
